# MicroRNAs expressed during normal wound healing and their associated pathways: A systematic review and bioinformatics analysis

**Morgana Lüdtke Azevedo**[1☯]*, **Roberta Giorgi Silveira**[2☯], **Fernanda Nedel**[2‡], **Rafael Guerra Lund**[1‡]*

**1** Graduated Program in Biochemistry and Bioprospecting, Federal University of Pelotas, Pelotas, RS, Brazil,
**2** Graduated Program in Health and Behavior, Catholic University of Pelotas, Pelotas, RS, Brazil

☯ These authors contributed equally to this work.
‡ FN and RGL also contributed equally to this work.
* morganaludtke@gmail.com (MLA); rafael.lund@gmail.com (RGL)

## Abstract

MicroRNAs (miRNAs) are responsible for regulating gene expression post-transcriptionally. Are involved in several biological processes, such as wound healing. Understanding the miRNAs involved in this process is fundamental for the development of new therapies. So, due to the need to understand the role of these molecules, we aimed systematically review the literature in order to identify which miRNAs are involved in the wound healing and determine, through bioinformatics analysis, which signaling pathways are associated with these miRNAs. An electronic search was performed in the following databases: National Library of Medicine National Institutes of Health (PubMed), Science Direct, Scifinder, Scopus and Web of Science, using the descriptors: "(microRNA [MeSH])" and "(skin [MeSH])" and "(wound healing [MeSH])". After the search, two independent and previously calibrated reviewers selected the articles that analyzed the expression pattern of miRNAs in wound healing in *in vivo* studies, using the software Zotero bibliography manager. Following, bioinformatic analysis was performed using the software DIANA Tools, mirPath v.3 and the data was interpreted. The bioinformatics analysis revealed that on the day 1 there were 13 union pathways, eight of which were statistically significant. Still on the day 1, among the miRNAs that had a decrease in their expression, 12 of 17 union pathways found were statistically significant. On the day 5, among the miRNAs with an increase in expression, 16 union pathways were found, 12 of which were statistically significant. Finally, among the miRNAs with decreased expression, 11 of 15 union pathways found were statistically significant. Although it has been found substantial heterogeneity in the studies, with this systematic review, it was possible to study the panorama of miRNAs that may be altered in the wound healing. The present review summarizes existing evidence of miRNAs associated to wound healing, and these findings can contribute to new therapeutic approaches.

**Data Availability Statement:** All relevant data are within the paper.

**Funding:** The authors received no specific funding for this work.

**Competing interests:** The authors have declared that no competing interests exist.

## 1. Introduction

Wound healing is a biological process extremely complex and able to recover the barrier function of the skin. It requires the synchronization of several cell types beyond the interaction between cytokines and growth factors. It is composed of four different phases: hemostasis, inflammation, proliferation, and remodeling. Each one of these phases is important so that the process can occur properly. However, during the proliferative phase occurs a cascade of events of major importance for the process as a whole, such as re-epithelization, angiogenesis, and wound closure [1].

The events associated to wound healing can be affected by a variety of agents, changing the wound bed environment. Excessive wound healing–hypertrophic scar and keloid–and chronic wound are some of the sequels of impaired wound healing [2, 3]. Indeed, this process is highly complex and consists of several steps, being more susceptible to a fault. Since wound healing does not occur normally, a chronic wound may be the most recurrent consequence [4]. Nowadays, chronic non-healing wounds can affect millions of patients each year, resulting in higher morbidity and mortality of these patients [5]. Furthermore, the issue related to chronic wounds is not limited only to ulcers but is also associated with other challenges to be overcome, such as infectious and ischemic wounds. It is also known the treatment of chronic non-healing wounds remains a stretch goal on the subject of complexity and prevalence [6]. Considering this, it is clear the need for a therapeutic approach able of overcoming the problem of the complexity of this wound type in terms of treatment.

At the same time, several studies have reported miRNA as a potential therapeutic target for various diseases or conditions, from wound healing to cancer [7–9]. MiRNAs are small non-coding RNA molecules of approximately 22 nucleotides and are responsible for the regulation of gene expression at the post-transcriptional level [10, 11]. They are involved in an assorted of biological processes, from embryonic development to main cell functions, such as proliferation, differentiation, and apoptosis. The miRNA biogenesis and mechanism of action are known only about two decades ago and its molecular mechanism and is not yet totally elucidated [12, 13]. The miRNA research is clearly expanding, mainly in the last five years, because increasingly more studies have elucidated some points of miRNA molecular mechanisms. Moreover, as the miRNAs are important regulators of gene expression, may be promising targets for the development of biomarkers and can help in the development of a therapy [14].

Although there are still some questions about miRNA to be clarified, it is currently known that it is a key part of the entire epigenetic machinery acting as an important regulator of gene expression [15]. The epigenetic plays an important role in all the processes that occur in living organisms and can be used to explain several features of diseases and biological events, such as their pathway or late onset and end [16]. However, miRNAs are not only a part of the epigenetic machinery but are also can modify epigenetically by DNA methylation and histone modification just like another protein-coding gene [17, 18]. These changes in the miRNA expression pattern occur also during the wound healing, and as reported several miRNAs can be found decreased, including the members of the miR-200 family [19], or increased like miR-31, miR-33 and miR-196, among others [20–23]. Also was reported that certain miRNAs, such as miR-21, can mitigate possible aging-associated wound healing failures [24].

Therefore, comprehension of the miRNAs role in wound healing, as well as where and how it acts, can be applied in identifying the pathways involved in the process in order to bring new possibilities of molecular target therapies to the defects that can occur during wound healing. Further, considering the lack of studies about the panorama of miRNAs and their signaling pathways, mainly related to wound healing, we aimed to systematically review the available literature for the purpose of identifying which miRNAs are associated with the wound healing phases and their associated pathways using bioinformatics analysis.

## 2. Material and methods

### 2.1 Registration and review questions

The current systematic review is reported in accordance with the Preferred Items for Systematic Review and Meta-Analysis (PRISMA) guidelines. Due to the study design's nature, the protocol was registered in the Open Science Framework and is available at the following link (https://osf.io/xp4tf/).

The review questions were: (1) Which miRNAs have been related to the wound healing process? (2) In which pathways do these miRNAs act?

### 2.2 Inclusion and exclusion criteria

The inclusion criterion for the articles was *in vivo* animal studies that analyzed the miRNA expression patterns in the wound healing process. The choice to use only *in vivo* animal studies is due to the similarity found in the methodology of these studies, reducing possible confounding variables. Only studies published in English were included. The exclusion criteria were congress summary, book section, literature reviews, hypothesis articles, opinion articles, methodological approaches, commentaries, previews, expert opinions, letters, news, patents, studies unrelated to miRNA and wound healing, and studies with confounding factors. Besides that, articles that were not written in English and not fully available were also excluded.

### 2.3 Search strategy

The electronic strategy was carried out without initial date restriction up to and including May 2020 in the following databases: PubMed, Science Direct, Scifinder, Scopus, and Web of Science. In this case, the Google Scholar database was not used due to the fact that it presents low precision in recently established themes, such as miRNAs [25, 26]. The search strategy was conducted using the following terms: "(MicroRNAs) AND (Skin) AND (Wound Healing)". No language was applied in the search.

### 2.4 Study selection

The search results have been exported to the Zotero bibliography manager. Initially, duplicate records were excluded. Titles, abstracts, and study methodologies were screened based on the inclusion and exclusion criteria by two blinded and independent reviewers (MLA and RGS). All records were compared and in case of disagreement, a consensus was reached by discussion. When consensus was not achieved, a different reviewer decided if the article should be included (RGL).

### 2.5 Data extraction

Data were extracted and tabulated independently by two reviewers (MLA and RGS) in an Excel spreadsheet (Microsoft Corporation, Redmond, WA, USA) to be submitted for descriptive analysis. Cases of disagreement were handled as described above.

### 2.6 Bioinformatics analysis

After extracting the data, the miRNAs that showed different expression patterns (higher or lower expression) and were repeated on days 1 and 5 during the wound healing process were separately inserted, using the software DIANA Toll mirPath v.3. The inclusion criterion for bioinformatics analysis were the miRNAs that were described more than one time at different times analyzed in the selected studies, and the exclusion criterion was the miRNAs described

just one time in the selected studies beyond the miRNAs do not index in mirPath v.3. So, the miRNAs that were not recognized by the software were disregarded in this analyze.

The analyses were performed in real-time using the Kyoto Encyclopedia of Genes and Genoma (KEGG) selecting the murine specie to investigate the miRNAs. The interactions dataset chosen for all miRNAs was TarBase v7.0 and in the advanced statistics options, False Discovery Rate (FDR) correction was chosen. The Fisher's Exact Test was used with a p-value threshold of 0.05. After inserting all miRNAs into the software, the pathway associated were observed and tabulated, specifying its p-value and its relationship to the wound healing process.

### 2.7 Quality assessment

To assess the quality of the studies included in this systematic review, the Review Manager 5.4.1 software was used. The checklist was composed of three domains: wound healing assay, miRNA analysis, and the presence of a control group (uninjured skin). Two independent researchers (MLA and RGS) assessed the quality of the studies based on criteria previously established. In cases of disagreement were discussed until a consensus was reached, and when a consensus was not obtained, a third reviewer participated in the discussion (RGL).

## 3. Results

The initial search amounted to 3,227 records. After the removal of duplicates, a total of 2,370 articles remained for the title and abstract screening. So, based on the title and abstract, 915 articles remained for full-text reading screening. After reading the full text, were excluded 847 articles, remaining 68 articles satisfy the inclusion criteria. These of 68 articles, 40 were excluded for not performing the analysis *in vivo* animal. A total of 28 articles remained, and 16 of these ended up being excluded for showing possible confounding variables, such as studies that used some type of treatment or transgenic animal. In other words, studies did not analyze wound healing under regular conditions. Finally, remained 12 articles that present viable data to analyze, as shown in the PRISMA flowchart for the study selection process (Fig 1).

Among the 12 selected studies, it is clear that the country of publication of the most recurrent articles is the USA, totaling 6 of the 12 articles, followed by China with 4 of these publications. Regarding the year of publication, the oldest article date from 2012, demonstrating how recent the miRNAs studies, especially in relation to wound healing, is recent. These results are also described and illustrated in the figure below (Fig 2).

Table 1 refers to the main methodologies used in the 12 selected articles. Concerning the animal model used, the choice was unanimous. Absolutely all articles used mice as an animal model, there were a few studies that used rats; however, these ended up being excluded due to eligibility criteria. The most frequent strains were C57BL/6, C57BL/6J, and Balb-C. Interestingly, the SKH-1 strain, which is hairless mice in the skin, was reported in only one article. Almost half of the studies specified the gender of their animals–female (n = 3) and male (n = 4)–the other half did not specify. Most animals were between 8–10 weeks old. The more frequent methodology applied for the induction of the lesion was full-thickness wound by biopsy punch in 100% of the articles. The recurrent induction site of the lesion was on the dorsal skin, and the lesions were sized between 3–6 mm.

As shown in S1 Table, the main techniques used to analyze the expression of miRNA were RT-qPCR, reported by most articles, and microarray. Some studies combined the two methodologies. The total amount of miRNA that had its expression analyzed varied according to the days of analysis. Among the most recurrent analysis times that were related to the wound healing phases, three days stood out: day 1, day 3, and day 5. However, the relationship between the down and up expression on day 3 was verified by only one of the selected articles (Aunin

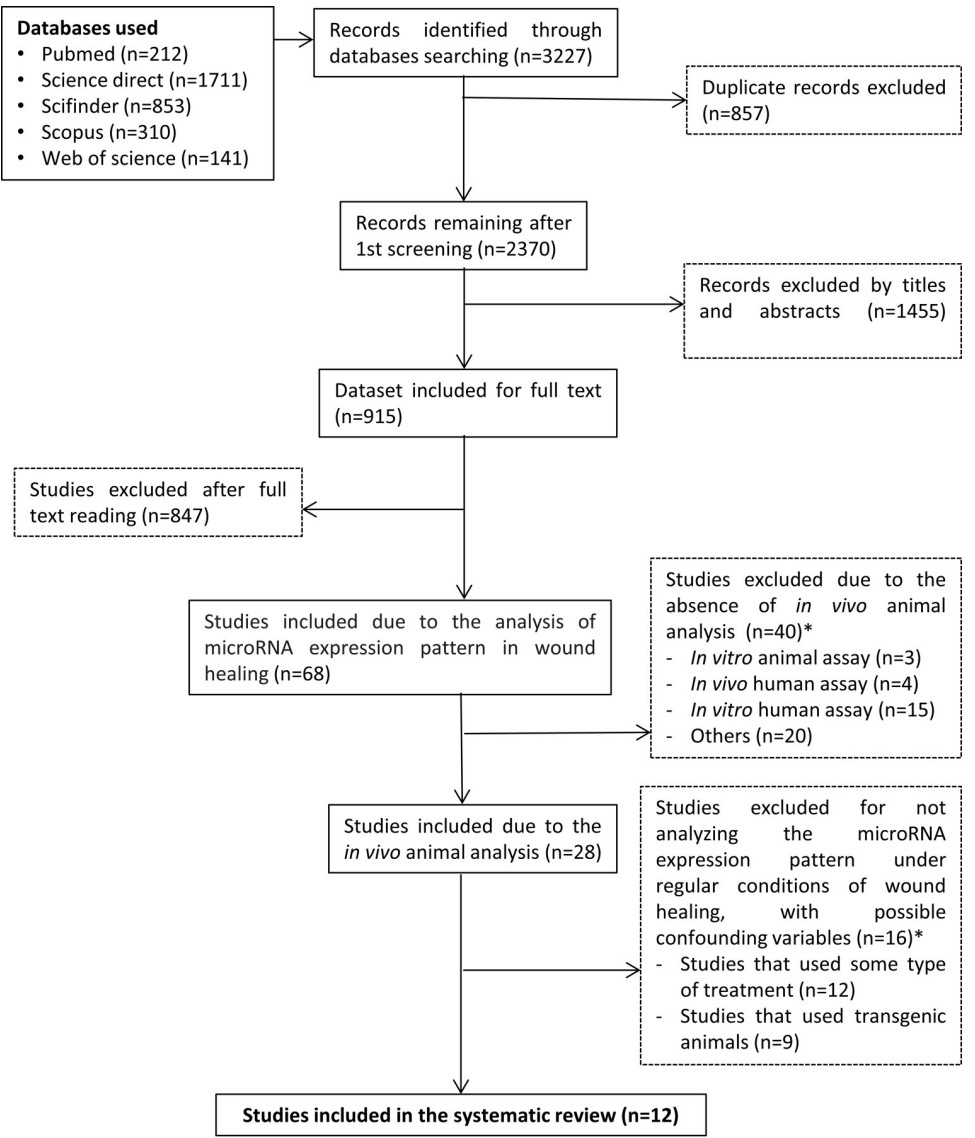

Databases used
- Pubmed (n=212)
- Science direct (n=1711)
- Scifinder (n=853)
- Scopus (n=310)
- Web of science (n=141)

Records identified through databases searching (n=3227)

Duplicate records excluded (n=857)

Records remaining after 1st screening (n=2370)

Records excluded by titles and abstracts (n=1455)

Dataset included for full text (n=915)

Studies excluded after full text reading (n=847)

Studies included due to the analysis of microRNA expression pattern in wound healing (n=68)

Studies excluded due to the absence of *in vivo* animal analysis (n=40)*
- *In vitro* animal assay (n=3)
- *In vivo* human assay (n=4)
- *In vitro* human assay (n=15)
- Others (n=20)

Studies included due to the *in vivo* animal analysis (n=28)

Studies excluded for not analyzing the microRNA expression pattern under regular conditions of wound healing, with possible confounding variables (n=16)*
- Studies that used some type of treatment (n=12)
- Studies that used transgenic animals (n=9)

Studies included in the systematic review (n=12)

* A study could have fulfilled more than one criterion

**Fig 1. Flow diagram of study selection.** Flow diagram of the study selection process according to PRISMA Statement.
* A Study could have fulfilled more than one criterion.

et al., 2017). The other articles found alteration of the expression in only one of these two patterns and not concurrently (Chan et al., 2012 (a), Chan et al., 2012 (b) and Zao et al., 2020 found alterations in just decreased expression. While, Shi et al., 2018 and Wang et al, 2019 found alterations in just increased expression). So the bioinformatics analysis was performed according to the data obtained on days 1 and 5.

On day 1, a total of 273 miRNAs were analyzed, and just two of these were not affected, in other words, do not have alteration in their expression pattern. Among these that showed some alteration in expression pattern, 153 miRNAs had an increase in their expression, while 118 miRNAs, decreased. On day 5, the time of analysis with the highest amount of analyzed

## Publication Country

China: (4)

Germany: (1)

United Kingdom: (1)

United States: (6)

## 12 selected studies

## Publication Year

2012: (3) 2013: (1)

2014: (1) 2016: (1)

2017: (1) 2018: (1)

2019: (3) 2020: (1)

**Fig 2. Correlation between publication countries and years of selected articles.** Correlation between publication countries and years of selected articles.

miRNA, the not affected ones totaled 5 miRNAs, while those with increased and decreased expression were 192 and 130, respectively. At last, considering these two days of analysis of the 12 selected articles, a total of 600 miRNAs had their expression pattern analyzed during the wound healing process. The most frequently found miRNAs on days 1 and 5 that were analyzed by the bioinformatic are described in Table 3.

The bioinformatics analysis discloses that on the day 1, among the miRNAs that had their expression increased, there were three intersection pathways, but none with a statistically significant difference ($p \leq 0.05$) and 13 union pathways, eight of which were statistically significant. In the miRNAs that had a decrease in their expression, 17 union pathways were found, 12 of which were statistically significant, and no one intersection pathway was found. On the day 5, in the miRNAs with an increase in expression, 16 union pathways were found, 12 of which were statistically significant, and also no one intersection pathway was found. Among the miRNAs with decreased expression, 15 union pathways were found, 11 of which were statistically significant, and no one intersection pathway was found (Tables 2–5), and signaling pathways related to the wound healing process are highlighted in bold.

**Table 1. Correlation of the main methodological approaches found in the selected articles.**

| Author, Year | Animal conditions | | | | Wound procedures | | |
| --- | --- | --- | --- | --- | --- | --- | --- |
| | Animal | Strain | Sex | Age | Injury induction site | Injury type | Injury size |
| Aunin et al., 2017 | Mice | - | - | 8 weeks | Back skin | Full-thickness wound by biopsy punch | 3 mm |
| | | | | 2 years | Back skin | Full-thickness wound by biopsy punch | 3 mm |
| Chan et al., 2012a | Mice | C57BL/6 | Male | 8–10 weeks | Dorsal skin | 2 or 4 full-thickness excisional wounds by biopsy punch | 6 mm |
| Chan et al., 2012b | Mice | C57BL/6 | Male | - | Dorsal skin | 2 full-thickness wound | 8x16 mm |
| | Mice | C57BL/6 | Male | - | Dorsal skin | 2 or 4 full-thickness excisional wounds by biopsy punch | 6 mm |
| Chen et al., 2019 | Mice | Balb/c | Female | 8–10 weeks | Dorsal skin | 2 full-thickness incisional wounds using a pair of scissors | 10 mm |
| | | | | | | 2 full-thickness excisional wounds by biopsy punch | 5 mm |
| | | | | | Anterior of the hard palate | 3 incisional wounds using a scalpel blade | 50 mm |
| Etich et al., 2017 | Mice | C57BL/6N | - | 8–10 weeks | Back skin | Full-thickness wounds by biopsy punch | 6 mm |
| Jin and Chung, 2018 | Mice | SKH-1 | Female | 8 weeks | Dorsal skin | 2 full-thickness wounds by biopsy punch | 3.5 mm |
| Shi et al., 2018 | Mice | - / K14-Cre | - | 7–8 weeks | Dorsal skin | Full-thickness wounds by biopsy punch | 6 mm |
| Simões et al., 2019 | Mice | Balb/c | Female | 8–10 weeks | Dorsal skin | Full-thickness wounds by biopsy punch | 5mm |
| | | | | | Hard palate from maxilla | Full-thickness with a pair of forceps | - |
| | | | | | Dorsal skin | 2 full-thickness excisional wounds by biopsy punch | 5 mm |
| van Solingen et al., 2014 | Mice | B6.Cg-Mirn155<sup>tm1.1Rsky</sup>/J | - | 10–12 weeks | - | 2 full-thickness by biopsy punch | 6 mm |
| | | C57BL | - | 10–12 weeks | - | 2 full-thickness by biopsy punch | 6 mm |
| Wang et al., 2012 | Mice | C57BL | Male | Adult | Back | Full-thickness skin excision | 10 mm |
| Wang et al., 2019 | Mice | C57BL (H-2b) | Male | Adult | Back | 2 full-thickness by biopsy punch | 6 mm |
| | | B6.Cg-Mirn155<sup>tm1.1Rsky</sup>/J | Male | 6–8 weeks | Back | 2 full-thickness by biopsy punch | 6 mm |
| Zhao et al., 2020 | Mice | C57BL/6J | - | 8–12 weeks | Back | 2 circular full-thickness excisional wounds | 6 mm |
| | | C57BL/6J | - | 8–12 weeks | Back | 2 circular full-thickness excisional wounds | 6 mm |
| | | - | Male and female | - | Back | 2 circular full-thickness excisional wounds | 6 mm |

Main findings regarding the methodological approaches of the studies include animal and strain, injury induction site, injury size, and wound procedure.

The miRNAs and their associated pathways also are illustrated in Fig 3. In the Venn diagrams (Fig 4) it is possible to observe the signaling pathways that overlap on both days (1 and 5), as well as those that occur exclusively on either day 1 or day 5.

Based on the quality assessment of the 12 studies, the three pre-established domains proved to be adequate (Fig 5). The requirement to perform the miRNA analysis in addition to the presence of a control group were the domains that presented insufficient or absent definitions according to pre-established domains.

**Table 2. Statistically significant union pathways refer to the miRNAs with an increased expression on day 1.**

| Pathway | p-value | Target genes |
|---|---|---|
| Fatty acid biosynthesis | <1e-325 | 2 |
| Fatty acid metabolism | 6.165574e-09 | 4 |
| **Steroid biosynthesis** | **6.921681e-08** | **3** |
| Central carbon metabolism in cancer | 2.532531e-05 | 9 |
| Caffeine metabolism | 2.639553e-05 | 2 |
| Proteoglycans in cancer | 0.009315449 | 8 |
| Carbohydrate digestion and absorption | 0.01156247 | 4 |
| Fat digestion and absorption | 0.01410009 | 2 |
| **Complement and coagulation cascades** | **0.04045149** | **7** |
| **Thyroid hormone and signaling pathway** | **0.04320222** | **11** |
| **HIF-1 signaling pathway** | **0.04440003** | **7** |
| Renal cell carcinoma | 0.04873515 | 7 |
| Viral carcinogenesis | 0.0496533 | 11 |

## 4. Discussion

In this systematic review, based on the results obtained, can be observed a panorama of altered miRNAs during the wound healing process and their pathways associated (Table 2).

Recent studies have often reported the importance of the role of miRNA in several cellular processes, including wound healing. These studies claim that during the process the miRNA expression pattern can alternate according to the day, suggesting its regulatory function in these cases [27–29]. In our results, it was possible to observe these patterns on different days of wound healing. on each of the days analyzed, essentially all the analyzed miRNAs suffered some alteration in their expression pattern. The miRNAs found with some alterations were quite varied, considering all analysis times.

**Table 3. Statistically significant union pathways refer to the miRNAs with a decreased expression on day 1.**

| Pathway | p-value | Target genes |
|---|---|---|
| Prion diseases | 6.72018e-13 | 7 |
| **Hippo signaling pathway** | **4.282252e-09** | **1** |
| Proteoglycans in cancer | 2.17414e-07 | 59 |
| Caffeine metabolism | 1.547323e-05 | 2 |
| Renal cell carcinoma | 9.472664e-05 | 17 |
| **Steroid biosynthesis** | **0.0001140205** | **5** |
| Lysine degradation | 0.00122778 | 18 |
| **Adherens junctions** | **0.001282294** | **20** |
| **TGF-beta signaling pathway** | **0.00177966** | **13** |
| **FoxO signaling pathway** | **0.003806427** | **47** |
| **Endocytosis** | **0.004723869** | **55** |
| **N-Glycan biosynthesis** | **0.0115209** | **15** |
| Drug metabolism–other enzymes | 0.01724572 | 6 |
| Protein processing in endoplasmatic reticulum | 0.0386971 | 55 |
| Cicardian entrainment | 0.04087073 | 1 |
| Morphine addiction | 0.04087073 | 1 |
| Pancreatic cancer | 0.04101645 | 23 |

**Table 4. Statistically significant union pathways refer to the miRNAs with an increased expression on day 5.**

| Pathway | p-value | Target genes |
|---|---|---|
| Fatty acid biosynthesis | <1e-325 | 2 |
| Fatty acid metabolism | 4.440892e-16 | 11 |
| Renal cell carcinoma | 1.630683e-08 | 23 |
| **Steroid biosynthesis** | **3.491662e-07** | **5** |
| **TGF-beta signaling pathway** | **0.0001216682** | **13** |
| Lysine degradation | 0.0003168722 | 10 |
| **Thyroid hormone signaling pathway** | **0.0005841115** | **25** |
| **FoxO signaling pathway** | **0.0007019681** | **31** |
| Central carbon metabolism in cancer | 0.003639735 | 6 |
| Proteoglycans in cancer | 0.01120219 | 19 |
| **HIF-1 signaling pathway** | **0.01131124** | **17** |
| Viral carcinogenesis | 0.01320044 | 26 |
| Bacterial invasion of epithelial cells | 0.0176033 | 17 |
| RNA degradation | 0.0310473 | 10 |
| **Regulation of actin cytoskeleton** | **0.03361088** | **37** |
| Axon guidance | 0.04726244 | 25 |

On day 1, among the miRNAs with an increase in their expression (mmu-miR-223-3p and mmu-miR-34c-5p), already it is possible to observe the correlation with wound healing. Evidence had shown that mmu-miR-223-3p can ameliorates vascular endothelial lesions through the IL6ST and STAT3 signaling pathways [23, 30]. Furthermore, members of the miR-223 family have been associated as important regulators in the inflammatory process that occurs during early phases of wound healing, which justifies its increase in expression on day 1, when the inflammatory process of tissue repair occurs. Also, the miR-223 family is effective in increasing the activation of neutrophils after episodes of bacterial infections, and consequently, improving the course of the healing process [31, 32]. The miR-34 family also seems to be

**Table 5. Statistically significant union pathways refer to the miRNAs with a decreased expression on day 5.**

| Pathway | p-value | Target genes |
|---|---|---|
| **Hippo signaling pathway** | 2.384482e-11 | 1 |
| Caffeine metabolism | 8.713539e-08 | 2 |
| Proteoglycans in cancer | 7.888269e-06 | 32 |
| Drug metabolism–other enzymes | 0.0001989493 | 6 |
| **Steroid biosynthesis** | 0.0003114483 | 2 |
| Cicardian entrainment | 0.0006915614 | 1 |
| Morphine addiction | 0.0006915614 | 1 |
| Lysine degradation | 0.001772916 | 12 |
| Metabolism of xenobiotics by cytochrome P450 | 0.00374839 | 4 |
| **Endocytosis** | 0.005002083 | 34 |
| **ErbB signaling pathway** | 0.008312138 | 18 |
| **MAPK signaling pathway** | 0.01457244 | 39 |
| Phosphatidylinositol signaling system | 0.01930813 | 16 |
| Colorectal cancer | 0.02517332 | 14 |
| Protein processing in the endoplasmic reticulum | 0.04973187 | 29 |

Correlation of binding pathways observed in miRNAs with increased and decreased expression and their respective target genes.

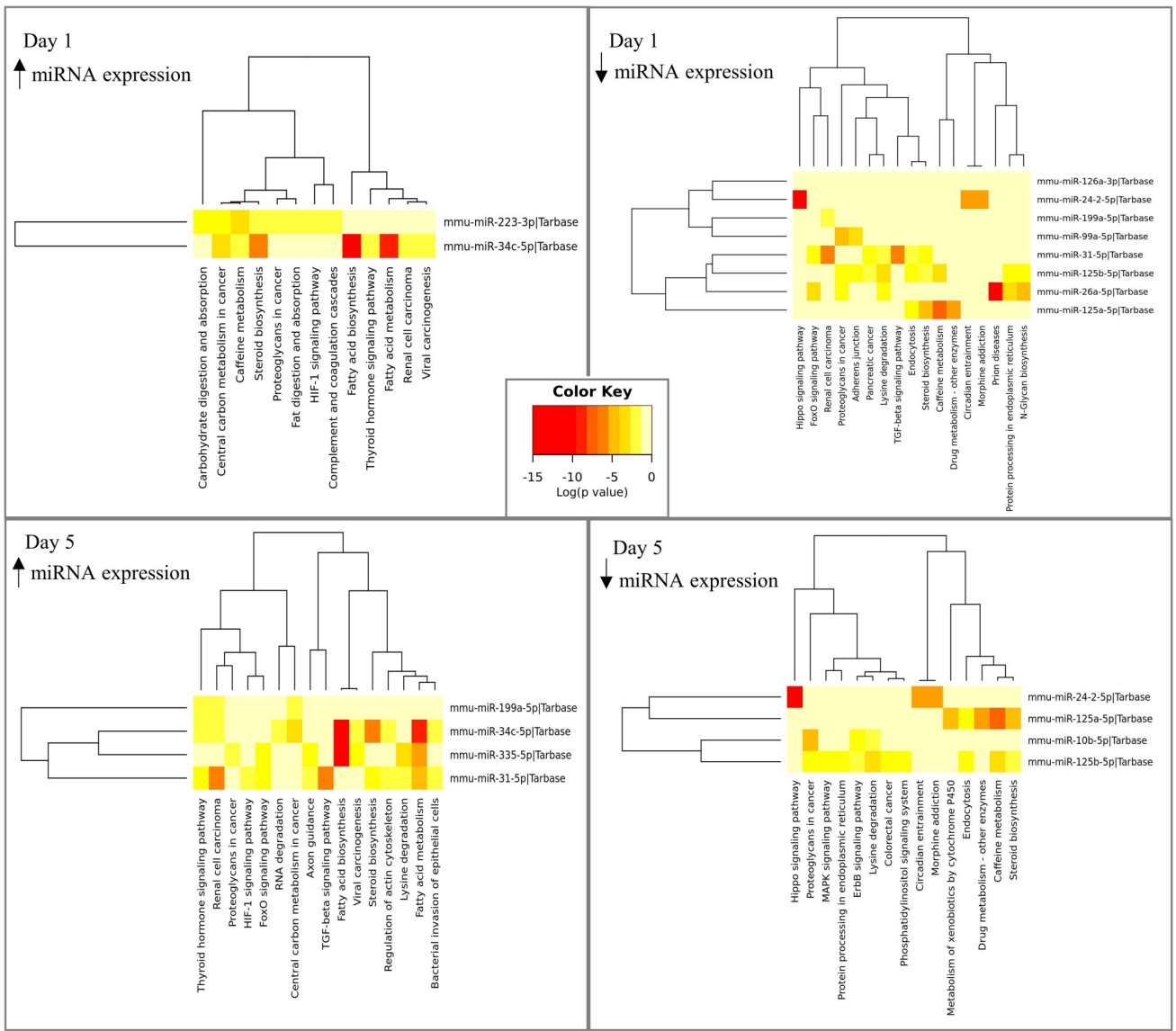

**Fig 3. Analysis of the miRNAs expression.** Heatmap of KEGG pathways referring to increased or decreased expression of miRNAs at different times of analysis.

related to wound healing; it has been reported that several family members are up-regulated in epidermal keratinocytes during wound healing. In addition, it being able to improve the inflammatory process occurring during healing due to the release of inflammatory cytokines [33, 34].

The miR-31 family is also closely related to the wound healing process, these miRNAs can regulate keratinocytes proliferation, differentiation, and migration through the regulation of the signaling pathways NF-κB, RAS/MAPK, Notch, and cytokines [35, 36]. The miR-31-5p was found with expression decreased on day 1 and increased on day 5. Already described for its important role in cell migration, the miR-199a-5p was found decreased on both days [37], but also there is evidence that miR-199a-5p can, in a negative way, regulate the angiogenic responses through responses by directly targeting ETS proto-oncogene 1 and transcription factor (ETS-1) [38]. The first miRNAs associated with inflammatory response were miR-132 and

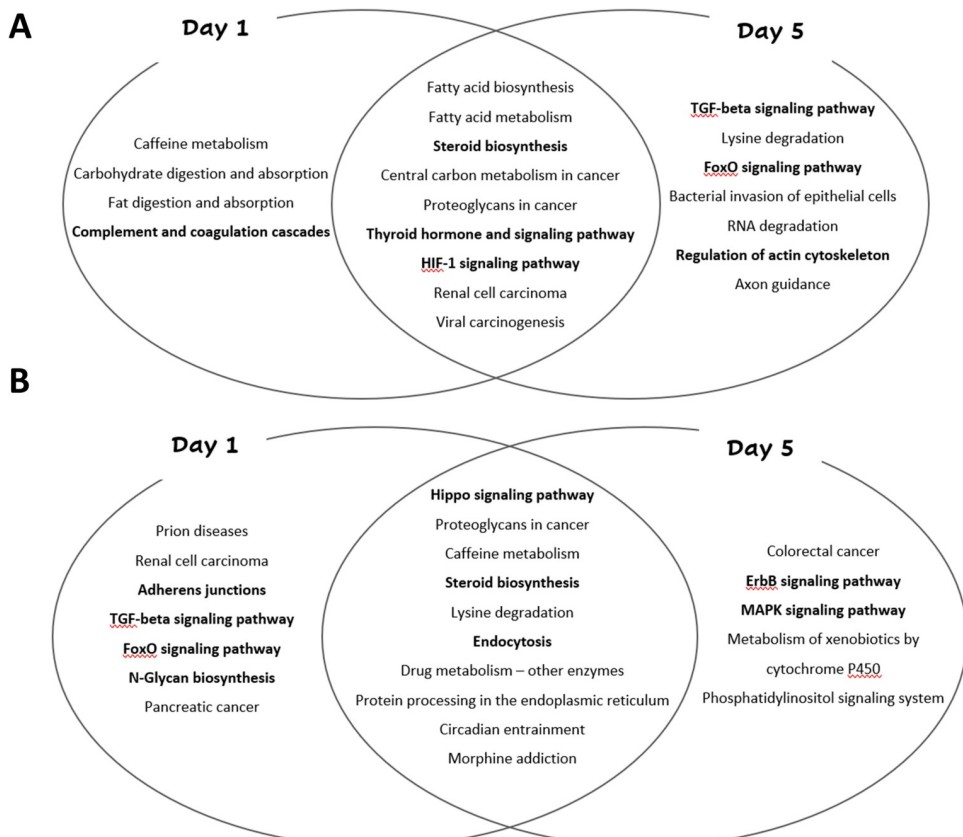

**Fig 4. (A). Analysis of significant union pathways overlapping, and non-overlapping.** Venn diagram representing the signaling pathways overlapping, and non-overlapping refer to the miRNAs with an increased expression on days 1 and 5. **(B). Analysis of significant union pathways overlapping, and non-overlapping.** Venn diagram representing the signaling pathways overlapping, and non-overlapping refer to the miRNAs with a decreased expression on days 1 and 5.

miR-125b, the expression was induced in a monocytic cell line treated with lipopolysaccharide [39–41]. However, the miR-125a-5p and miR-125b-5p were found with a decreased expression on both days. The angiogenesis also is regulated by the miRNAs. For instance, some studies already related the downregulation of miR-199a-5p expression in the dermis and endothelial tissue during the wound healing process. Also, by targeting an angiogenesis-related transcription factor and its mediator, the miR-199a-5p can, negatively, regulate the angiogenic response of dermal microvascular endothelial cells in humans. In mice with homozygous deletions in the ETS-1 gene, was related to an impaired of angiogenesis, insufficient formation of granulation tissue, and compromised wound closure [41–43]. The miRNA-199a-5p was found with an expression decreased on day 1, but not on day 5.

In our bioinformatics analysis, we investigated the associated pathways highlighting the statistically significant and related to wound healing. Among them, the thyroid hormone signaling pathway (p = 0.04) was identified. This pathway can be related to several biological processes by regulating gene expression. The thyroid hormone, for instance, already was described as one of the most potent stimulators of growth and metabolic rate, it can induce the angiogenesis through the increase of bGFG mRNA expression via the integrin αvβ3/PKD/HDAC5 signaling pathway [44, 45]. In a culture of human keratinocytes, the exogenous thyroid human stimulated the expression of keratin genes, these genes are responsible for 30% of

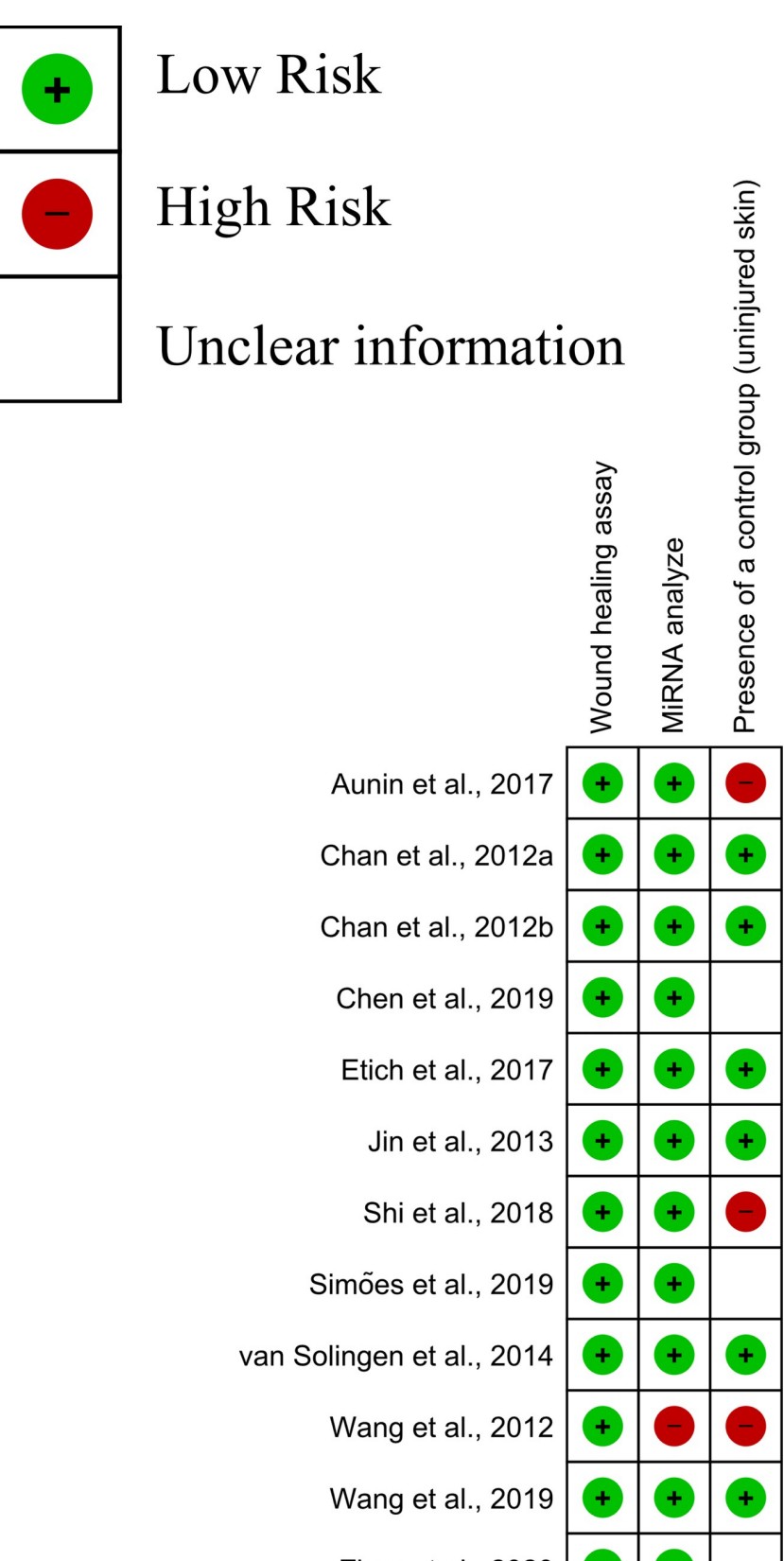

**Fig 5. Analysis of risk of bias of the selected studies.** Risk of bias of each included study in the domains: wound healing assay, miRNA analysis, and presence of a control group.

the protein of the epidermis, and there is clear evidence about the relation to keratin genes and wound healing specific phases [46]. Another representative pathway associated with miRNAs with an increased expression is the Hypoxia Inducible Factor-1 (HFI-1) signaling pathway (p = 0.04). Several studies already related the role of HIF-1 in wound healing, contributing to cell migration and division under hypoxic conditions, beyond the growth factor release and extracellular matrix [47, 48].

The forkhead box O (FoxO) signaling pathway was also found among the miRNAs with an increased expression (p = 0.0007) targeting more than 30 genes. The FoxO family is constituted of transcription factors responsible for regulating gene expression in several cellular events and biological processes, such as apoptosis, cell-cycle control, oxidative stress resistance and wound healing stimulation [49]. Also, the members of this family are recruited for keratinocyte mobilization and migration due to their ability to regulate, positively, transforming growth factor-beta (TGF-β1) expression. The TGF-β1 exerts effects on wound healing through immune modulation, cell proliferation, migration and differentiation regulation, and extracellular matrix production [50]. The TGF-β1 signaling pathway was also found associated to wound healing (p = 0.0001). There are three isoforms (TGF-β1, TGF-β2 and TGF-β3) and all of these appears to exert effects on wound healing through the SMAD pathway, mainly. The TGF-β1 is more frequently related to scarless wound healing formation, whereas the TGF-β3 already was observed in fibrotic scarring [51]. These findings corroborate the TGF-β1 decrease described in chronic non-healing wounds [52]. In general, the TGF-β family can play a role he wound healing through inflammation regulation, fibroblast proliferation, angiogenesis simulation, and deposition and remodeling of the extracellular matrix [53].

Among the signaling pathways associated with the miRNAs with decreased expression and related to the wound healing process, was found the steroid biosynthesis pathway (p = 0.0001). This signaling pathway can be related to wound healing in many ways; glucocorticoids (GC) are able to inhibit wound healing through their membranous glucocorticoid receptor. This receptor via activation of the Wnt-like 6 PLC/PKC signaling cascade interferes with the keratinocytes migration and, consequently, on wound closure [54]. The Wnt signaling pathway already was, frequently, related to several aspects of skin development and physiology. Also, in cases of Wnt pathway reduction, the regenerative capacities and abilities are impaired. This pathway regulates the β-catenin activation, and this process appears to be one of the several inflammatory responses to injury [54, 55]. Although the Wnt signaling pathway in the inflammatory response is not yet very understood, some evidence, through the observation of gene Wnt5 increased expression in patients with severe sepsis, suggests β-catenin-independent Wnt signaling may be a proinflammatory stimulus, a key event for the wound healing process [56].

Cell adhesion, mediated by adherens junctions, is crucial and closely related to the wound healing process. The adherens junction signaling pathway (p = 0. 001) plays an important role in cell plasticity, providing both cell-cell adhesion and fast cell-cell contact remodeling during several biological processes, such as wound healing [57]. The adherens junctions are also a key target of endocytosis during wound healing; the endocytosis signaling pathway (p = 0.004), also found in miRNAs with decreased expression, provides the endocytic remodeling of adherens junctions. This cell adhesion is required to control the actin assembly on the wound edge increasing the speed of wound closure [58]. There is some evidence suggesting that most cell-cell adhesion proteins can be modified by N-Glycans, increasing the probability of the occurrence of defects in the formation of the protective barrier and cell differentiation and adhesion

[59]. The N-Glycans biosynthesis signaling pathway (p = 0.01), and these glycans are clearly involved in processes responsible for regulating the terminal differentiation products in keratinocytes [60].

In addition, were identified two signaling pathways among the miRNAs with decreased expression involving the protein kinase. The ErbB (epidermal growth factor receptor) signaling pathway (p = 0.08) is a family of receptor tyrosine kinases (RTKs) responsible for binding extracellular growth factor to intracellular pathways, to regulate some biological responses–cell proliferation, differentiation, and motility [61]. These growth factors, acting via RTKs, are able to control different cell types in skin wound healing, particularly macrophages and neutrophils. For this reason, the aberrant expression of the growth factors or their receptors is related to difficult wound healing [62]. Cell proliferation is also regulated for the mitogen-activated protein kinase (MAPK) signaling pathway (p = 0.01). Through more than 18 target genes, the activation of the MAPK pathway, mainly the extracellular signal-regulated kinase (ERK), is the most important regulator of several cell types' migration. The ERK/MAPK pathway can be activated by skin damage and this activation has a strong effect on keratinocyte migration [63–66].

Ultimately, this systematic review has limitations that need to be highlighted. Different strains of mice were used to observe the miRNA pattern expression during wound healing, although there are certain strains most frequently used, their differences must be considered so that the results are properly interpreted. In different strains, beside the miRNA pattern expression, the signaling pathways also can be altered [67, 68]. Another variable that can induce confusion and doubts is the analysis time. Wound healing is a complex and extensive process, in other words, does not occur in a fixed time, but in three or four phases taking to days to months [1, 3]. Thus, it is necessary to define the most adequate analysis time to obtain the most reliable data possible, according to the studied species and the biological processes that the objective is to study. Nonetheless, in some cases, such heterogeneity may be important, for example, to identify new pathways that would not be related otherwise.

Hence, the results found in this systematic review and bioinformatics analysis contribute to the identification of miRNAs altered during the wound healing process, as well as the associated signaling pathways. Furthermore, they can be used as a study tool for the next works that aim to relate miRNAs to wound healing, in the search for new potential biomarker targets, whether for diagnosis or therapy.

## 5. Conclusion

In conclusion, the results of our systematic review demonstrated that some miRNAs are altered during the wound healing process. The bioinformatics analysis revealed that on day 1, among the miRNAs with increased and decreased expression, there are 20 union pathways that were statistically significant. And on day 5, we can observe a similar amount, totalizing 23 union pathways statistically significant. Most miRNAs identified in our study play a role in wound healing through regulating, mainly, cell proliferation and differentiation, by several signaling pathways. It is worth noting the limitation we found in the selected studies regarding their significant heterogeneity, which can be explained by the differences in the target populations—in this case, the wound healing is observed in different strains of mice -, and timing of outcome measurements. In this sense, even though there is a diversity in the studies found, this heterogeneity can be important in order to identify new pathways that would probably not be considered in another manner, as in studies with low heterogeneity. But even then, the results we present help to better understand the complex network of miRNAs, as well as their role in the healing of wounds. With this systematic review, it was possible to study the

panorama of miRNAs that may be altered in the wound healing, understanding which miR-NAs and its respective signaling pathways may be involved in the wound healing process. The present review summarizes existing evidence of miRNAs associated to wound healing, and these findings can contribute to new therapeutic approaches.

## Supporting information

**S1 Checklist. PRISMA 2020 checklist.**
(DOCX)

**S1 Table. Correlation of results regarding the expression pattern of miRNAs at different times of analysis.**
(DOCX)

## Author Contributions

**Data curation:** Morgana Lüdtke Azevedo, Roberta Giorgi Silveira, Fernanda Nedel, Rafael Guerra Lund.

**Formal analysis:** Morgana Lüdtke Azevedo, Roberta Giorgi Silveira.

**Investigation:** Morgana Lüdtke Azevedo, Roberta Giorgi Silveira, Fernanda Nedel, Rafael Guerra Lund.

**Methodology:** Morgana Lüdtke Azevedo, Roberta Giorgi Silveira, Fernanda Nedel, Rafael Guerra Lund.

**Project administration:** Morgana Lüdtke Azevedo.

**Resources:** Roberta Giorgi Silveira.

**Software:** Morgana Lüdtke Azevedo, Roberta Giorgi Silveira.

**Supervision:** Morgana Lüdtke Azevedo, Roberta Giorgi Silveira, Fernanda Nedel, Rafael Guerra Lund.

**Validation:** Morgana Lüdtke Azevedo, Roberta Giorgi Silveira, Fernanda Nedel, Rafael Guerra Lund.

**Visualization:** Morgana Lüdtke Azevedo, Roberta Giorgi Silveira, Fernanda Nedel, Rafael Guerra Lund.

**Writing – original draft:** Morgana Lüdtke Azevedo, Roberta Giorgi Silveira.

**Writing – review & editing:** Morgana Lüdtke Azevedo.

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
