## [Decision Letter · Decision Letter 0]

23 Dec 2021

PONE-D-21-30551MicroRNAs expressed during wound healing and their associated pathways: a systematic review and bioinformatics analysisPLOS ONE

Dear Dr. Azevedo,

Thank you for submitting your manuscript to PLOS ONE. After careful consideration, we feel that it has merit but does not fully meet PLOS ONE’s publication criteria as it currently stands. Therefore, we invite you to submit a revised version of the manuscript that addresses the points raised during the review process.

We look forward to receiving your revised manuscript.

Kind regards,

Andrea Caporali, PhD

Academic Editor

PLOS ONE

Journal Requirements:

2. Please report your search date in Search strategy section.

The authors thank the financial support from the Brazilian National Council for Scientific and Technological Development (CNPq).

5. Please amend the manuscript submission data (via Edit Submission) to include author ROBERTA GIORGI SILVEIRA, FERNANDA NEDEL and RAFAEL GUERRA LUND.

Reviewers' comments:

Reviewer's Responses to Questions

**Comments to the Author**

1. Is the manuscript technically sound, and do the data support the conclusions?

Reviewer #1: Yes

Reviewer #2: Yes

Reviewer #3: Partly

2. Has the statistical analysis been performed appropriately and rigorously? 

Reviewer #1: Yes

Reviewer #2: I Don't Know

Reviewer #3: Yes

3. Have the authors made all data underlying the findings in their manuscript fully available?

Reviewer #1: Yes

Reviewer #2: Yes

Reviewer #3: Yes

4. Is the manuscript presented in an intelligible fashion and written in standard English?

Reviewer #1: No

Reviewer #2: Yes

Reviewer #3: No

5. Review Comments to the Author

Reviewer #1: The authors present a timely and interesting analysis of miRNAs associated with wound healing in mouse skin. As the work is focused on physiological healing in healthy mice, perhaps the title needs to reflect this: MicroRNAs expressed during normal wound healing...

The manuscript needs editing for English language usage.

RESULTS

It is not entirely clear how the authors went from 2,370 articles with titles and abstracts to 915 articles for full text reading. If this this based on >1,400 articles either not being in English or not freely available, this needs to be made clear.

The authors state that day 3 was analysed by only one of the selected articles (page 10, line 188,189). But there are at least 6 studies in Table 2 that have a day 3 collection time: Aunin et al 2017, Chan et al 2012a, 2012b, Shi et al 2018, Wang et al 2019; Zhao et al 2020. So the authors need to remove or clarify that comment about day 3 analysis.

Looking at the day 1 and day 5 pathways for miRNA with increased expression on day 1 and day 5 (Tables 3 and 5), it might be useful to show the overlap between the pathways (e.g. using a Venn diagram). This would shed light on pathways that feature at both times points and delineate them from those that are limited to day 1 or day 5. Same argument goes for data in Tables 4 and 6, for miRNAs with reduced expression.

There were 273 miRNAs for day 1 and 327 for day 5 (p21, line 196ff). Conceivably all the 273 day 1 miRNAs featured in the 327 at day 5, but it would be good to make this clear or else generate a Venn diagram showing the overlap between the two datasets.

DISCUSSION

While the comments link miRNAs and targets to wound healing, there needs to be deeper consideration of the relationships between the miRNAs and the transcripts. Consider ErbB and MAPK signalling (p31,lines 338 ff). What makes these pathways noteworthy is not just that there are 18 and 39 target genes, respectively (Table 6) but rather the fact that they feature in the union pathways for miRNAs with DECREASED expression – that is, the targets themselves are likely to be upregulated to support cell proliferation given that the expression of their miRNA regulators is reduced. The same logic applies to most of the pathways considered.

The above also raises further questions: should additional/supplementary Results tables be generated showing the names of the target genes for the “interesting” pathways – “interesting” pathways being the ones considered in the Discussion?

In a similar vein, as presented, we cannot tell which miRNAs are associated with a given pathway. I am not sure if there is a way that allows this to be done easily but that seems to be the missing link between Table 2 and Tables 3-6. If not for all the targets, such insight might at least be relevant for those targets at the overlap/intersection of the day 1 and day 5 data that I mentioned in the Results.

Minor

The Simoes et al data in particular (in Table 2) have multiple entries that are not mouse miRNAs (i.e. not prefixed with mmu). This needs some explanation.

The papers in Tables 1 and 2, cited with author names, need the corresponding reference number from the bibliography for ease of cross-referencing.

Tables 3-5 have some pathways in bold. It is not clear what the bold font is supposed to signify, if anything.

Page 21,line 204: Table 2 rather than Table 4?

I could see Figure titles but the manuscript does not seem to have included figure legends.

Reviewer #2: Dear Author

1. MicroRNAs expressed during wound healing and their associated pathways: a systematic review and bioinformatics analysis require English corrections.

2. author has to include the miRs inhibition in wound healing and explain it in separate analysis

Reviewer #3: The systematic review by Azevedo et al. reported numerous miRNAs to regulate different phases of wound healing, in particular the inflammation and proliferation phases. The outcome of this review maybe used to guide preclinical and clinical studies on the role of miRNA in wound healing. In my view, the manuscript cannot be published in its present form.

Major comments:

1. Numerous typographical and grammatical errors have been observed. The English, formulation of certain sentences, and paragraphs need huge improvement to provide readers better understandability of the text.

2. Improvise abstract to provide gist of the manuscript.

3. Although the authors mentioned to include all articles involving in vivo animal studies, but only mice studies were included in table 1. Were there no studies using rats or rabbit?

4. Additionally, some points that need further clarification:

- In table 1, difference between back skin and dorsal skin?

- Sources of miRNAs, were the miRNAs extracted from peripheral blood or wound tissues?

5. It would be beneficial if the authors include these in the discussion:

- Comparison between miRNA signatures between day 1 and day 5.

- Relate the miRNA signatures or union pathways to the inflammatory and proliferation phases of wound healing.

6. PLOS authors have the option to publish the peer review history of their article (what does this mean?). If published, this will include your full peer review and any attached files.

Reviewer #1: **Yes: **Kehinde Ross

Reviewer #2: No

Reviewer #3: No

---

## [Author Response · Author response to Decision Letter 0]

15 Jun 2022

Journal Requirements:

Authors’ response:

We have modified the level of our manuscript titles and subtitles, and also, we added title to each to the figures, as required by the PLOS ONE. Thank you for your suggestions.

2. Please report your search date in Search strategy section.

Authors’ response:

Thank you very much for your suggestion. We added our search date in Search strategy section.

Authors’ response:

We corrected the mistake in the ‘Funding Information’ and ‘Financial Disclosure’. Thank you for letting us know this mistake.

The authors thank the financial support from the Brazilian National Council for Scientific and Technological Development (CNPq).

Authors’ response:

Thank you very much for your suggestion. We corrected the Acknowledgement section of our manuscript. The ‘Funding Statement’ information are already correct.

5. Please amend the manuscript submission data (via Edit Submission) to include author ROBERTA GIORGI SILVEIRA, FERNANDA NEDEL and RAFAEL GUERRA LUND.

Authors’ response:

The other authors’ names have been included. Thank you for letting us know this mistake.

Authors’ response:

The subtitles have been included. Thank you for letting us know this mistake.

Reviewers' comments:

Reviewer's Responses to Questions

Comments to the Author

1. Is the manuscript technically sound, and do the data support the conclusions? 

Reviewer #1: Yes

Reviewer #2: Yes

Reviewer #3: Partly

Author’s comment:

Thank you for your comments. Our study was conducted rigorously with adequate controls and experimental design.

2. Has the statistical analysis been performed appropriately and rigorously?

Reviewer #1: Yes

Reviewer #2: I Don't Know

Reviewer #3: Yes

Author’s comment:

The statistical analysis of our study has been performed adequately to support their conclusion. Thank you very much for your comments.

3. Have the authors made all data underlying the findings in their manuscript fully available?

Reviewer #1: Yes

Reviewer #2: Yes

Reviewer #3: Yes

Author’s comment:

Thank you very much for all your comments.

4. Is the manuscript presented in an intelligible fashion and written in standard English?

Reviewer #1: No

Reviewer #2: Yes

Reviewer #3: No

Author’s comment:

Thank you very much for all your comments. Our study has been corrected by an English teacher and the grammatical errors has been corrected.

5. Review Comments to the Author

Reviewer #1:

- The manuscript needs editing for English language usage.

Our study has been corrected by an English teacher and the grammatical errors has been corrected.

- RESULTS

It is not entirely clear how the authors went from 2,370 articles with titles and abstracts to 915 articles for full text reading. If this this based on >1,400 articles either not being in English or not freely available, this needs to be made clear.

Thank you very much for your comment. The explanation of why the articles went from 2370 to 915 are explained in the following lines: 156 and 157. After removing the duplicates, 2370 articles remained for the title and abstract screening, resulting in 915 articles.

The authors state that day 3 was analysed by only one of the selected articles (page 10, line 188,189). But there are at least 6 studies in Table 2 that have a day 3 collection time: Aunin et al 2017, Chan et al 2012a, 2012b, Shi et al 2018, Wang et al 2019; Zhao et al 2020. So the authors need to remove or clarify that comment about day 3 analysis.

Thank you very much for this comment. Actually, the sentence is poorly structured, because the day 3 was analyzed by more the one study. But, just in one we can see results that compare the up and down regulation. We really appreciate this comment. The authors had already corrected the error in the article.

- DISCUSSION

While the comments link miRNAs and targets to wound healing, there needs to be deeper consideration of the relationships between the miRNAs and the transcripts. Consider ErbB and MAPK signalling (p31,lines 338 ff). What makes these pathways noteworthy is not just that there are 18 and 39 target genes, respectively (Table 6) but rather the fact that they feature in the union pathways for miRNAs with DECREASED expression – that is, the targets themselves are likely to be upregulated to support cell proliferation given that the expression of their miRNA regulators is reduced. The same logic applies to most of the pathways considered.

Thank you very much for this comment. The authors have already reformulated in the article.

The above also raises further questions: should additional/supplementary Results tables be generated showing the names of the target genes for the “interesting” pathways – “interesting” pathways being the ones considered in the Discussion?

Yes, the interesting pathways are those discussed in the discussion that have statistical significance.

I could see Figure titles but the manuscript does not seem to have included figure legends.

The authors already add figure legends.

Reviewer #2:

1. MicroRNAs expressed during wound healing and their associated pathways: a systematic review and bioinformatics analysis require English corrections.

Our study has been corrected by an English teacher and the grammatical errors has been corrected.

2. Author has to include the miRs inhibition in wound healing and explain it in separate analysis.

The inhibition of miRNAs in wound healing was included in the tables.

Reviewer #3: The systematic review by Azevedo et al. reported numerous miRNAs to regulate different phases of wound healing, in particular the inflammation and proliferation phases. The outcome of this review maybe used to guide preclinical and clinical studies on the role of miRNA in wound healing. In my view, the manuscript cannot be published in its present form.

Major comments:

1. Numerous typographical and grammatical errors have been observed. The English, formulation of certain sentences, and paragraphs need huge improvement to provide readers better understandability of the text.

Thank you very much for this comment. We reviewed our study with the help of a professional English teacher and corrected any perceived errors.

2. Although the authors mentioned to include all articles involving in vivo animal studies, but only mice studies were included in table 1. Were there no studies using rats or rabbit?

No, none of the studies performed the analyzes in rats or rabbit.

3. Additionally, some points that need further clarification:

- In table 1, difference between back skin and dorsal skin?

According to the figures provided in the selected studies, there is no difference in “back skin” and “dorsal skin”. 

- Sources of miRNAs, were the miRNAs extracted from peripheral blood or wound tissues?

The miRNAs were extracted from wound tissues.

4. It would be beneficial if the authors include these in the discussion:

- Comparison between miRNA signatures between day 1 and day 5.

- Relate the miRNA signatures or union pathways to the inflammatory and proliferation phases of wound healing.

Thank you very much for all this comments. The suggestions were accepted.

5. . PLOS authors have the option to publish the peer review history of their article. If published, this will include your full peer review and any attached files.

Yes.

---

## [Decision Letter · Decision Letter 1]

18 Jul 2022

PONE-D-21-30551R1MicroRNAs expressed during wound healing and their associated pathways: a systematic review and bioinformatics analysisPLOS ONE

Dear Dr. Azevedo,

Thank you for submitting your manuscript to PLOS ONE. After careful consideration, we feel that it has merit but does not fully meet PLOS ONE’s publication criteria as it currently stands. Therefore, we invite you to submit a revised version of the manuscript that addresses the points raised during the review process.

We look forward to receiving your revised manuscript.

Kind regards,

Andrea Caporali, PhD

Section Editor

PLOS ONE

Reviewers' comments:

Reviewer's Responses to Questions

**Comments to the Author**

1. If the authors have adequately addressed your comments raised in a previous round of review and you feel that this manuscript is now acceptable for publication, you may indicate that here to bypass the “Comments to the Author” section, enter your conflict of interest statement in the “Confidential to Editor” section, and submit your "Accept" recommendation.

Reviewer #1: (No Response)

Reviewer #3: (No Response)

2. Is the manuscript technically sound, and do the data support the conclusions?

Reviewer #1: Yes

Reviewer #3: Partly

3. Has the statistical analysis been performed appropriately and rigorously? 

Reviewer #1: Yes

Reviewer #3: Yes

4. Have the authors made all data underlying the findings in their manuscript fully available?

Reviewer #1: Yes

Reviewer #3: Yes

5. Is the manuscript presented in an intelligible fashion and written in standard English?

Reviewer #1: No

Reviewer #3: No

6. Review Comments to the Author

Reviewer #1: While the authors have made some improvements, the work still needs substantial English language revision in my view.

Results: Perhaps "After exclusion of 1455 articles based on titles and abstract screening, 915 articles remained for the analysis" would make the point clearer.

Discussion: need for deeper considueration of the relationships between the miRNAs and their targets appears not to have been addressed.

Minor points raised appear not to be been addressed.

Reviewer #3: After comparing the original submission and the revised manuscript, the authors only made minimal changes to the subheadings and acknowledgement.

The manuscript still contain numerous typographical and grammatical errors, which the authors need to improve in order to provide readers better understandability of the text.

Additionally, the abstract needs to revise to provide the gist of the review.

7. PLOS authors have the option to publish the peer review history of their article (what does this mean?). If published, this will include your full peer review and any attached files.

Reviewer #1: **Yes: **Kehinde Ross

Reviewer #3: No

---

## [Decision Letter · Decision Letter 2]

25 Nov 2022

PONE-D-21-30551R2MicroRNAs expressed during wound healing and their associated pathways: a systematic review and bioinformatics analysisPLOS ONE

Dear Dr.Azevedo,

Thank you for submitting your manuscript to PLOS ONE. After careful consideration, we feel that it has merit but does not fully meet PLOS ONE’s publication criteria as it currently stands. Therefore, we invite you to submit a revised version of the manuscript that addresses the points raised during the review process.

We look forward to receiving your revised manuscript.

Kind regards,

Andrea Caporali, PhD

Section Editor

PLOS ONE

Journal Requirements:

Additional Editor Comments:

Please consider minor comments from reviewer 1 and perform the language editing of the text. 

Reviewers' comments:

Reviewer's Responses to Questions

**Comments to the Author**

1. If the authors have adequately addressed your comments raised in a previous round of review and you feel that this manuscript is now acceptable for publication, you may indicate that here to bypass the “Comments to the Author” section, enter your conflict of interest statement in the “Confidential to Editor” section, and submit your "Accept" recommendation.

Reviewer #1: (No Response)

Reviewer #3: All comments have been addressed

2. Is the manuscript technically sound, and do the data support the conclusions?

Reviewer #1: Partly

Reviewer #3: Yes

3. Has the statistical analysis been performed appropriately and rigorously? 

Reviewer #1: I Don't Know

Reviewer #3: N/A

4. Have the authors made all data underlying the findings in their manuscript fully available?

Reviewer #1: Yes

Reviewer #3: No

5. Is the manuscript presented in an intelligible fashion and written in standard English?

Reviewer #1: No

Reviewer #3: Yes

6. Review Comments to the Author

Reviewer #1: "However, the relationship between 191 the down and up expression on the day 3 was analyzed by only one of the selected articles". I can see two articles with both up and down regulated miRNA considered. Also, why some ters in tables 3-5 are bold remains unexplained. Issues with the language remain. I was going for "No recommendation" but that is not an option so either minor revision or reject but I will not look at this manuscript again.

Reviewer #3: (No Response)

7. PLOS authors have the option to publish the peer review history of their article (what does this mean?). If published, this will include your full peer review and any attached files.

Reviewer #1: **Yes: **Kehinde Ross

Reviewer #3: No

---

## [Author Response · Author response to Decision Letter 2]

18 Jan 2023

January 4th, 2023

Dear Dr. Emily Chenette,

Editor-in-Chief of PLOS ONE

We would like to thank the reviewers for their consideration, which certainly contributed to improving the overall quality of the manuscript. We have considered all of them in the revised version, and re-submit our manuscript entitled “MicroRNAs expressed during wound healing and their associated pathways: a systematic review and bioinformatics analysis” for your further consideration.

We have prepared point-by-point responses to the review’s comments, and these are indicated below in red font. Also, the changes made in the text were tracked.

We hope that all these changes meet the requirements of the Journal and make the manuscript acceptable for publication in PLOS ONE. Thank you for considering the revised version of our manuscript.

Reviewers' comments:

Reviewer's Responses to Questions

Comments to the Author

1. If the authors have adequately addressed your comments raised in a previous round of review and you feel that this manuscript is now acceptable for publication, you may indicate that here to bypass the “Comments to the Author” section, enter your conflict of interest statement in the “Confidential to Editor” section, and submit your "Accept" recommendation.

Reviewer #1: (No Response)

Reviewer #3: All comments have been addressed

Authors comments:

Thank you for your consideration. We really attempt to adequately address all reviewers' comments in a previous review round.

2. Is the manuscript technically sound, and do the data support the conclusions?

Reviewer #1: Partly

Reviewer #3: Yes

Authors comments:

Thank you for your consideration. We already attempt to modify the original manuscript version with technically sound scientific research data that supports the conclusion. Besides, the systematic review methodology was conducted rigorously, with appropriate controls, replication, and sample sizes.

3. Has the statistical analysis been performed appropriately and rigorously?

Reviewer #1: I Don't Know

Reviewer #3: N/A

Authors comments:

4. Have the authors made all data underlying the findings in their manuscript fully available?

Reviewer #1: Yes

Reviewer #3: No

Authors comments:

Thank you for your consideration. We attempt to make all data underlying the findings described in the manuscript fully available without restriction.

5. Is the manuscript presented in an intelligible fashion and written in standard English?

Reviewer #1: No

Reviewer #3: Yes

Authors comments:

Thank you for your consideration. In the current version of the manuscript, the language was corrected with the help of a professional English teacher.

6. Review Comments to the Author

Reviewer #1: "However, the relationship between 191 the down and up expression on the day 3 was analyzed by only one of the selected articles". I can see two articles with both up and down regulated miRNA considered. Also, why some ters in tables 3-5 are bold remains unexplained. Issues with the language remain. I was going for "No recommendation" but that is not an option so either minor revision or reject but I will not look at this manuscript again.

Authors comments:

Thank you for your consideration. In fact, the day 3 was analyzed by more than one article. However, the alteration of the two possible patterns of expression (down or up) was verified in only one of the articles (Aunin et al., 2017). We have improved the sentence containing this information so that the explanation is clearer.

The bold terms in tables 3-5 are the signaling pathways related to the wound healing process. In the current version of the manuscript, we inserted this information before the tables, and also highlighted the signaling pathways related to the wound healing process in table 6, which was missing.

In the current version of the manuscript, the language was corrected with the assistance of a professional English teacher.

Reviewer #3: (No Response)

7. PLOS authors have the option to publish the peer review history of their article (what does this mean?). If published, this will include your full peer review and any attached files.

We have no problem to publish the peer review history of our article.

---

## [Editor Report · Decision Letter 3]

5 Feb 2023

MicroRNAs expressed during wound healing and their associated pathways: a systematic review and bioinformatics analysis

PONE-D-21-30551R3

Dear Dr. Azevedo,

We’re pleased to inform you that your manuscript has been judged scientifically suitable for publication and will be formally accepted for publication once it meets all outstanding technical requirements.

Kind regards,

Andrea Caporali, PhD

Section Editor

PLOS ONE
---

## [Editor Report · Acceptance letter]

20 Feb 2023

PONE-D-21-30551R3 

MicroRNAs expressed during normal wound healing and their associated pathways: a systematic review and bioinformatics analysis 

Dear Dr. Azevedo:

I'm pleased to inform you that your manuscript has been deemed suitable for publication in PLOS ONE. Congratulations! Your manuscript is now with our production department. 

Kind regards, 

on behalf of

Dr Andrea Caporali 

Section Editor

PLOS ONE